# A Novel FFT_YOLOX Model for Underwater Precious Marine Product Detection

**Peng Wang [1], Zhipeng Yang [2], Hongshuai Pang [2], Tao Zhang [3] and Kewei Cai [2,*]**

1  College of Mechanical and Power Engineering, Dalian Ocean University, Dalian 116023, China; pwang0602@163.com
2  College of Information Engineering, Dalian Ocean University, Dalian 116023, China; yangzhipeng1101@126.com (Z.Y.); panghs@126.com (H.P.)
3  Dalian Tianzheng Industrial Co., Ltd., Dalian 116036, China; zht_3000@163.com
*  Correspondence: caikw0602@live.cn

**Featured Application: The proposed FFT_YOLOX model is a computational efficient and conceptually simple architecture to capture global information of images. This method could enhance the efficiency and accuracy of precious marine product detection in the underwater environment. Therefore, it can help the underwater robot to recognize and fish marine products automatically.**

**Abstract:** In recent years, the culture and fishing of precious marine product are heavily dependent on manual work, which is labor-intensive, high-cost and time-consuming. To address this issue, an underwater robot can be used to monitor the size of the marine products and fish the mature ones automatically. Automatic detection of marine products from underwater images is one of the most important steps in developing an underwater robot perceiving method. In the traditional detection model, the CNN based backbone suffers from the limited receptive field and hinders the modeling of long-range dependencies, due to the small kernel size. In this paper, a novel detection model FFT_YOLOX based on a modified YOLOX is proposed. Firstly, a unique FFT_Filter is presented, which is a computational efficient and conceptually simple architecture to capture global information of images. Then, a novel FFT_YOLOX model is introduced with fewer model parameters and FLOPs by replacing the standard $3 \times 3$ kernel in the original backbone of the YOLOX model with a FFT_Filter, for an underwater object detection vision task. Extensive experimental results demonstrate the effectiveness and generalization of the visual representation of our proposed FFT_YOLOX model.

**Keywords:** deep learning; FFT; precious marine product detection; underwater; YOLOX

## 1. Introduction

In recent years, precious marine product, including sea cucumbers, sea urchin and scallop, etc., have been widely noticed due to their use as food and folk medicines in communities across the world. The aquaculture of these products is growing rapidly with high economic benefits. Therefore, the fishing and culture of precious marine products are becoming much more significant. However, these fishing activities are mainly dependent on manual work, which is labor-intensive, high-cost and time-consuming. Furthermore, long-term underwater work is extremely threatening to human beings. With the rapid growth of underwater robot technology, automated fishing of sea cucumbers, sea urchin and scallop is becoming increasingly popular. An underwater robot can be used to monitor the size of marine products, and fish the mature ones automatically with high efficiency, labor saving.

Automatic detection of marine products from underwater images is one of the most important steps in developing an underwater robot perceiving method of previous marine products for fishing in the real world. In the literature, several image-processing methods for precious products, such as sea cucumbers, have been developed. Some specific features

of underwater objects have been considered and used for their detection. Separating the region of interest from the background based on edge features are proposed by [1–3]. These methods can achieve a good performance base on an appropriate threshold value of gray levels of objects and background. To acquire better results, further features are considered in detection methods [4–6], including shape, color, intensity and texture. In these traditional image processing approaches, the features used to recognize marine objects are hand-crafted, and the accuracy of detection is deeply related to the effectiveness of key feature extraction.

In recent years, deep learning (DL) has become increasingly popular. Convolutional neural networks (CNN) are considered to be the most representative DL method. CNNs have been widely used in a series of computer vision tasks, such as image classification [7–11], object detection [12–14], and semantic segmentation [15–18]. For object detection, several popular methods have been proposed, including two-stage and one-stage frameworks. A two-stage framework firstly carries out a region with a high probability of interest objects, then the proposal regions are used as the inputs of CNN for object recognition. The classical two-stage methods, e.g., R-CNN [19], R-FCN [20], Fast RCNN [21], and Faster RCNN [14], can obtain high accuracy, but suffer from complicated computation and slow inference time. A one-stage framework object detection method can address the previously proposed issues. The YOLO family [22–26] methods are the most popular one-stage detection methods, whilst several other underwater detection algorithms have also been proposed. For instance, Cai et al. [27] proposed a modified YOLO v3 model for fish detection, which changed the original backbone into MobileNet [28–30] v1 to decrease the parameters of its detection model and accelerate inference process. In [31], an underwater fish is detected online using a YOLO-based model. Recently, the YOLOX [26] is one of the most popular YOLO-series methods. It combines the region proposal process with object detection and proposes an anchor-free detection head for increased efficiency and reduced computation cost. Therefore, in this study we used the YOLOX as our basic detection framework.

In previous detection models, the backbone, which is used for feature extraction, is mainly reliant on CNNs, such as ResNets [32] and darknet [22–25]. In CNN architecture, the discrete convolutional filters (e.g., $3 \times 3$, $5 \times 5$ or $7 \times 7$ kernels) are used to capture spatial information of an image locally. Due to the small kernel size, it suffers from a limited receptive field which hinders the modeling of long-range dependencies. To address this issue, a unique backbone is proposed by replacing the $3 \times 3$ kernel in the original backbone of YOLOX with a conceptually simple, and computationally efficient, block, named an FFT_filter. A 2D discrete fast Fourier transform (FFT) [33] has been determined to be equivalent to a circular convolution, of which its size is the same as the feature map. Therefore, it can capture the global feature within the input feature map. The basic idea behind this architecture is to increase the receptive field of the FFT by learning the global interactions of spatial location.

In this study, three species of precious marine products, sea cucumber, sea urchin and scallop, were obtained as the research objects. A novel detection model based on a modified YOLOX is proposed. The contributions of this work are: (1) a unique FFT_filter is presented to replace the standard $3 \times 3$ kernel in the original backbone of YOLOX to capture global features and enlarge the receptive field; (2) a unified and modified YOLOX model, named FFT_YOLOX, is proposed for precious marine products detection in real underwater situation. Extensive experimental results demonstrate that the proposed detection method can outperform the original YOLOX model with fewer parameters.

The remainder of this paper is organized as follows: Section 2 presents an overview of YOLOX architecture and the 2D FFT method which are the basis of the proposed algorithm. Then, the FFT_filter is introduced and the main framework proposed. Section 3 presents the experiment results and discussion. Finally, the concluding remarks are illustrated in Section 4.

## 2. Methods

### 2.1. Overview of YOLOX

The YOLOX model is derived from the YOLO family, which is a one-stage object detection algorithm. The YOLOX framework mainly includes YOLOX-s, YOLOX-m, YOLOX-l and YOLOX-x models. Generally, YOLOX uses the modified CSPDarknet53 as its backbone for feature extraction, PANet as its neck for feature fusion, and a novel anchor-free algorithm as its detection head. Since it is the most popular and convenient one-stage detector, we selected it as our baseline. Furthermore, we chose YOLOX-s model to detect underwater objects, due to its low computation cost but competitive precision. In order to pursue better performance of precious marine product detection, we modified the backbone of the original YOLOX-s to have fewer model parameters but higher accuracy.

### 2.2. FFT_YOLOX

The framework of FFT_YOLOX is illustrated in Figure 1. We modified the original YOLOX-s to enlarge its receptive field and create the capability of capturing global features of an input image. In Figure 1, a CSP_FFT block is used to learn global interactions in a feature map. We replaced the 3 × 3 block by an FFT_Filter module in the original CSP block of the backbone of YOLOX-s. In the meantime, a 3 × 3 filter, which is not in the CSP block, was preserved to distill the information from the original image. Then, the proposed FFT_Filter could globally capture the features from a convolved map. A goal of this design was to create a global capture feature with a large receptive field. Furthermore, it was feasible to reduce model complexity with FFT_Filter module counterparts, to enhance the backbone of the YOLOX-s. Other modules were inherited from the original YOLOX-s detection framework. For instance, FPN and PANet were used as neck parts, whilst an anchor-free, decoupled YOLO detector enacted the detection head.

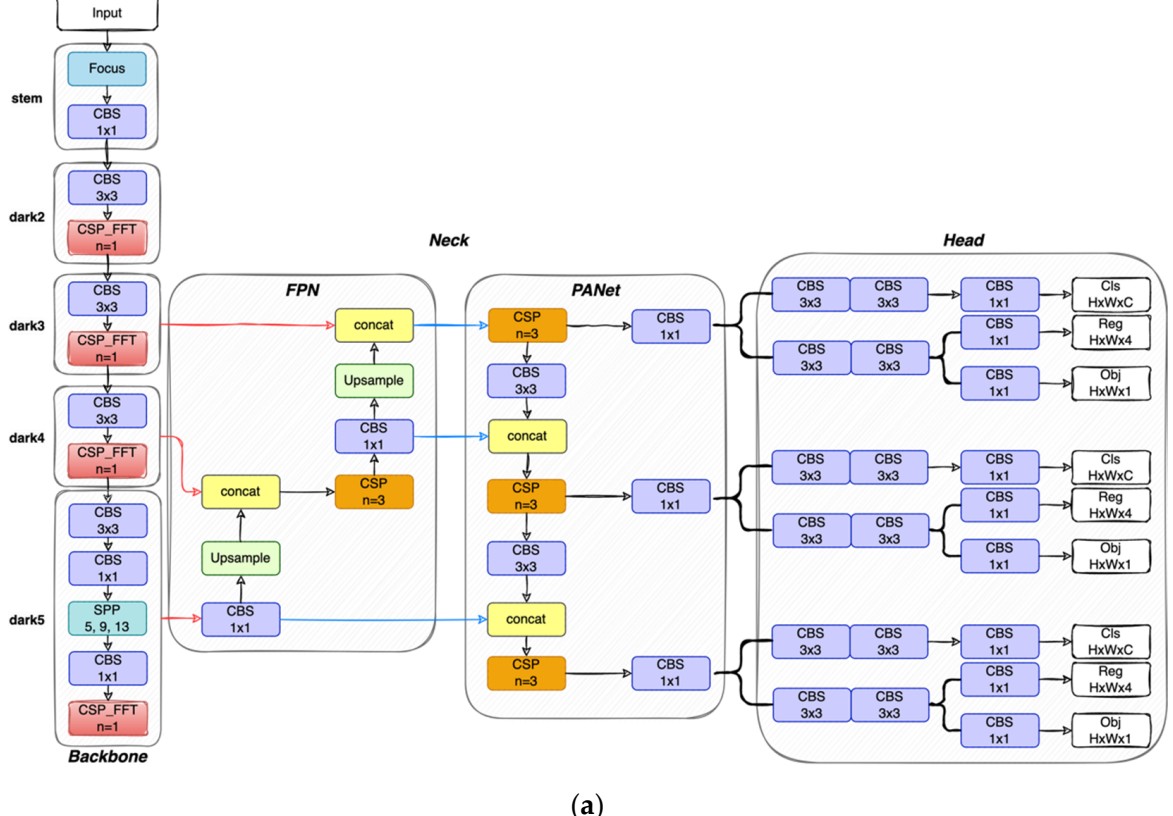

(**a**)

**Figure 1.** *Cont.*

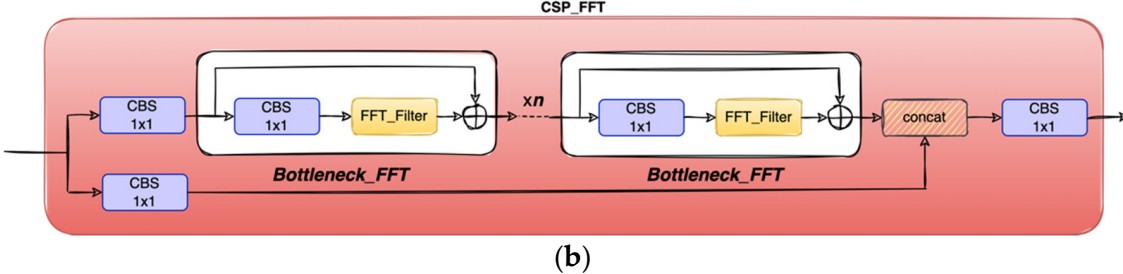

**(b)**

**Figure 1.** The framework of FFT_YOLOX, (**a**) the proposed modified YOLOX framework, (**b**) the proposed CSP_FFT block.

*2.3. FFT_Filter*

In this section, a brief review of an FFT_Filter module is introduced in Figure 2 for global feature extraction. To illustrate, the input feature map is set to $x \in \mathcal{R}^{H \times W \times C}$, where $H \times W$, $C$ represent the spatial resolution and number of channels, respectively. In FFT_Filter, a 2D FFT is performed to convert the feature map into spectral domain [34]:

$$\boldsymbol{X} = F[x] \in \boldsymbol{C}^{H \times W \times C} \tag{1}$$

$$X[u, v] = F(x[m, n]) = \Sigma_{m=0}^{M-1}\Sigma_{n=0}^{N-1} x[m,n]e^{-j2\pi(\frac{vm}{M} + \frac{un}{N})} \tag{2}$$

where, $F[x]$ denotes a Discrete Fourier Transform (DFT) operator, as illustrated in formula (2). $X[u, v]$ represents to the spectrum of the sequence $x[m, n]$ at frequencies $w_v = 2\pi\frac{vm}{M}$ and $w_u = 2\pi\frac{un}{M}$ in 2D domain. Inspired by the spectral convolution theorem [IF2-46], updating a single spectral domain value can globally affect all original data, which sheds light on design and the efficiency of the operator to capture global information in a feature map with a non-local receptive field. In the proposed FFT_Filter, a learnable matrix $\boldsymbol{FFT\_Kernel} \in \boldsymbol{C}^{H \times W \times C}$ is proposed to element-wise multiplicate with matrix $\boldsymbol{X}$ in the complex domain $\boldsymbol{C}^{H \times W \times C}$:

$$\hat{X} = \boldsymbol{FFT\_Kernel} \odot \boldsymbol{X} \tag{3}$$

where $\odot$ denotes the element multiplication. Additionally, Formula (3) is an efficient way to carry out a convolution with a global filter, of which size is $H \times W$. Then, an inverse DFT operator is used to revert the feature map into a real number field from its complex domain. Finally, the input feature map is merged with the inversed DFT result by an element add operator. In addition, to replace each $3 \times 3$ filter in the original CSPDarknet block in YOLOX-s, an average pooling is performed to satisfy the situation that stride is greater than 1.

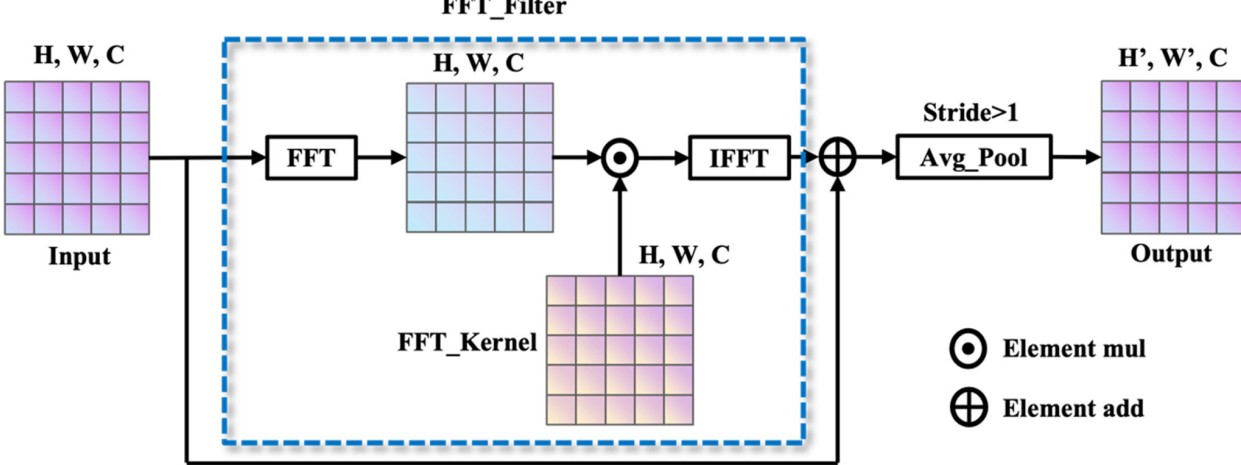

**Figure 2.** The architecture of FFT_filter.

## 3. Experiment and Discussion

Several experiments were conducted to demonstrate and evaluate the effectiveness of the proposed precious marine products detection model. The dataset is taken from underwater robot picking contest (UPRC) and the 0.5−0.95 mAP metric is adopted for evaluation.

### 3.1. Dataset

The proposed method is comprehensively evaluated on the dataset based on URPC. It offers a challenging object detection dataset, which contains a wide range of overlapping, blurred and occluded underwater creatures. We merged the URPC datasets across different years from 2018−2021 and deleted the most redundant images to establish a new underwater image dataset for model evaluation. This dataset consisted of 6328 images for training and 2187 for validation over four categories, including holothurian, echinus, scallop and starfish. Our proposed method was implemented on Pytorch 1.8. The hardware used to train and test the model was on an NVIDIA RTX3080Ti GPU.

### 3.2. Results and Comparison

We evaluated the proposed FFT_YOLOX for the object detection task on the merged URPC dataset. For this task, the original YOLOX-S model was adopted as the base detector. For fair comparison, we trained all models across 300 epochs by using SGD with weight decay 0.0005, momentum 0.9, and mini-batch of 8 on one 3080Ti GPU.

The detection results of FFT_YOLOX model are illustrated in Figure 3. From Figure 3, the effectiveness of the model is evaluated qualitatively. The most of precious marine products can be detected precisely, including small and indistinct objects which are hard to recognize even by human beings. Moreover, the qualitative comparison results of the original YOLOX-S and the proposed FFT_YOLOX model are shown in Figure 4. The proposed FFT_YOLOX model can detect more scallop and echinus from the difficult underwater environment, which the original YOLOX-S model cannot recognize. It owes to the capability of the global capturing feature of the FFT_Filter.

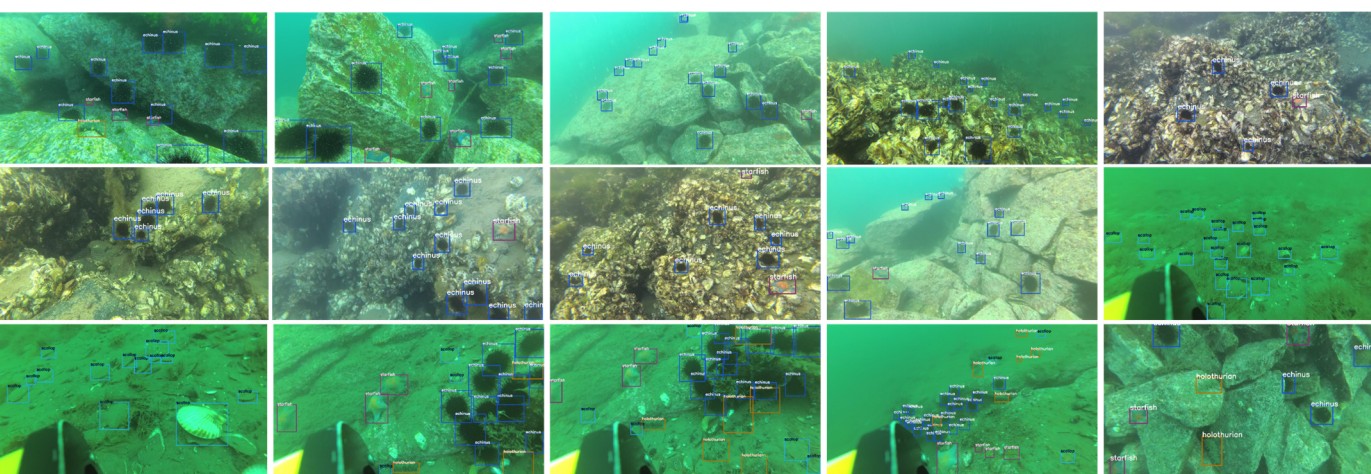

**Figure 3.** The detection results of FFT_YOLOX model.

For quantitative evaluation, the 0.5, 0.75, and 0.5−0.95 AP metrics are adopted for evaluation, and the AP of small, medium and large objects are also taken for performance comparisons. From Table 1, it can be seen that the proposed FFT_YOLOX model outperforms the original YOLOX-S across all AP metrics (The best results are illustrated with bold format). Moreover, each class AP (0.5−0.95) comparisons are shown in Table 2 (The best results are illustrated with bold format). The APs of holothurian, echinus, and starfish of the proposed model are higher than original YOLOX-S. It demonstrates that, due to the combination with global filters, the proposed FFT_Filter has stronger ability of feature ex-

traction with fewer model parameters and lower FLOPs than the original YOLOX-S model.

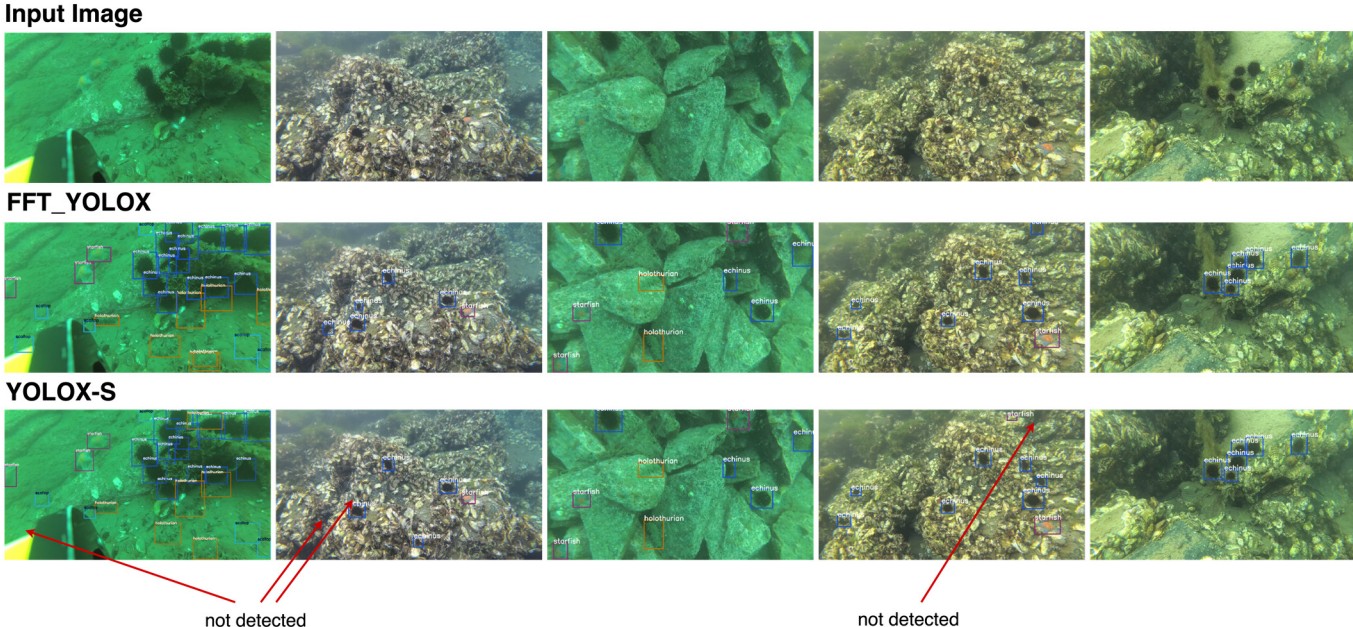

**Figure 4.** The qualitative comparison results of the original YOLOX-S and the proposed FFT_YO-LOX model.

**Table 1.** Performance comparisons on URPC dataset with YOLOX-S and FFT_YOLOX.

| Model | $AP_{0.5-0.95}$ | $AP_{0.5}$ | $AP_{0.75}$ | $AP_{small}$ | $AP_{medium}$ | $AP_{large}$ | Params | FLOPs |
|---|---|---|---|---|---|---|---|---|
| YOLOX-S | 0.480 | 0.829 | 0.506 | 0.207 | 0.418 | 0.531 | 8.94 M | 26.64 G |
| FFT_YOLOX | **0.483** | **0.832** | **0.510** | **0.242** | **0.421** | **0.533** | **8.49 M** | **25.18 G** |

**Table 2.** Each class AP (0.5−0.95) comparisons on URPC dataset with YOLOX-S and FFT_YOLOX.

| Model | Holothurian | Echinus | Scallop | Starfish | Params | FLOPs |
|---|---|---|---|---|---|---|
| YOLOX-S | 0.375 | 0.480 | **0.476** | 0.498 | 8.94 M | 26.64 G |
| FFT_YOLOX | **0.377** | **0.485** | 0.472 | **0.501** | **8.49 M** | **25.18 G** |

### 3.3. Ablation Study

We investigated how the designed FFT_Filter block in FFT_YOLOX influenced the whole performance of the detection model. Table 3 (The best results are illustrated with bold format) exhibits the performances in detail. The 0.5−0.95 AP metric of original YOLOX-S was taken as a baseline, demonstrating that the accuracy of 0.5−0.95 AP is 0.480 without FFT_Filter block. Then, we removed the FFT_Filter to replace the $3 \times 3$ block with the dark2-5 blocks of the YOLOX_S model. To illustrate, we designed four different models (exp2, exp3, exp4, exp5), which adds CSP_FFT block in dark2-5, dark3-5, dark4-5, and dark5 modules, respectively. As shown in Table 3, the proposed FFT_YOLOX (exp2) model, where CSP_FFT blocks were introduced in all dark2-dark5 modules of the original YOLOX-S model, had the highest accuracy of 0.5−0.95 AP with fewest model parameters and lowest FLOPs. Specifically, the accuracy of 0.5−0.95 AP of FFT_YOLOX (exp2) model, where only one CSP_FFT block was introduced as the dark5 module of the original YOLOX-S model, is also at a high level. This demonstrates that our proposed FFT_Filter enhanced the ability of global feature extraction even within a deep layer of a convolution network. Furthermore, for qualitative comparison, the detection results of different models are shown in Figure 5. Furthermore, to make the illustration of detection results be clearer, two important areas on the detected images are enlarged to highlight the comparison performances of different

models by using larger images, which are shown in Figure 6. From the results, it can be seen that the proposed FFT_YOLOX (exp2) model more accurately detected increased numbers of accurate precious marine products within the complex underwater environment.

**Table 3.** The detail performances of ablation study.

| Model | Dark 2 | Dark 3 | Dark 4 | Dark 5 | $AP_{0.5-0.95}$ | Params | FLOPs |
|---|---|---|---|---|---|---|---|
| YOLOX-S | - | - | - | - | 0.480 | 8.94 M | 26.64 G |
| FFT_YOLOX (exp2) | √ | √ | √ | √ | **0.483** | **8.49 M** | **25.18 G** |
| FFT_YOLOX (exp3) | - | √ | √ | √ | 0.481 | 8.50 M | 25.36 G |
| FFT_YOLOX (exp4) | - | - | √ | √ | 0.482 | 8.54 M | 25.59 G |
| FFT_YOLOX (exp5) | - | - | - | √ | 0.482 | 8.71 M | 26.46 G |

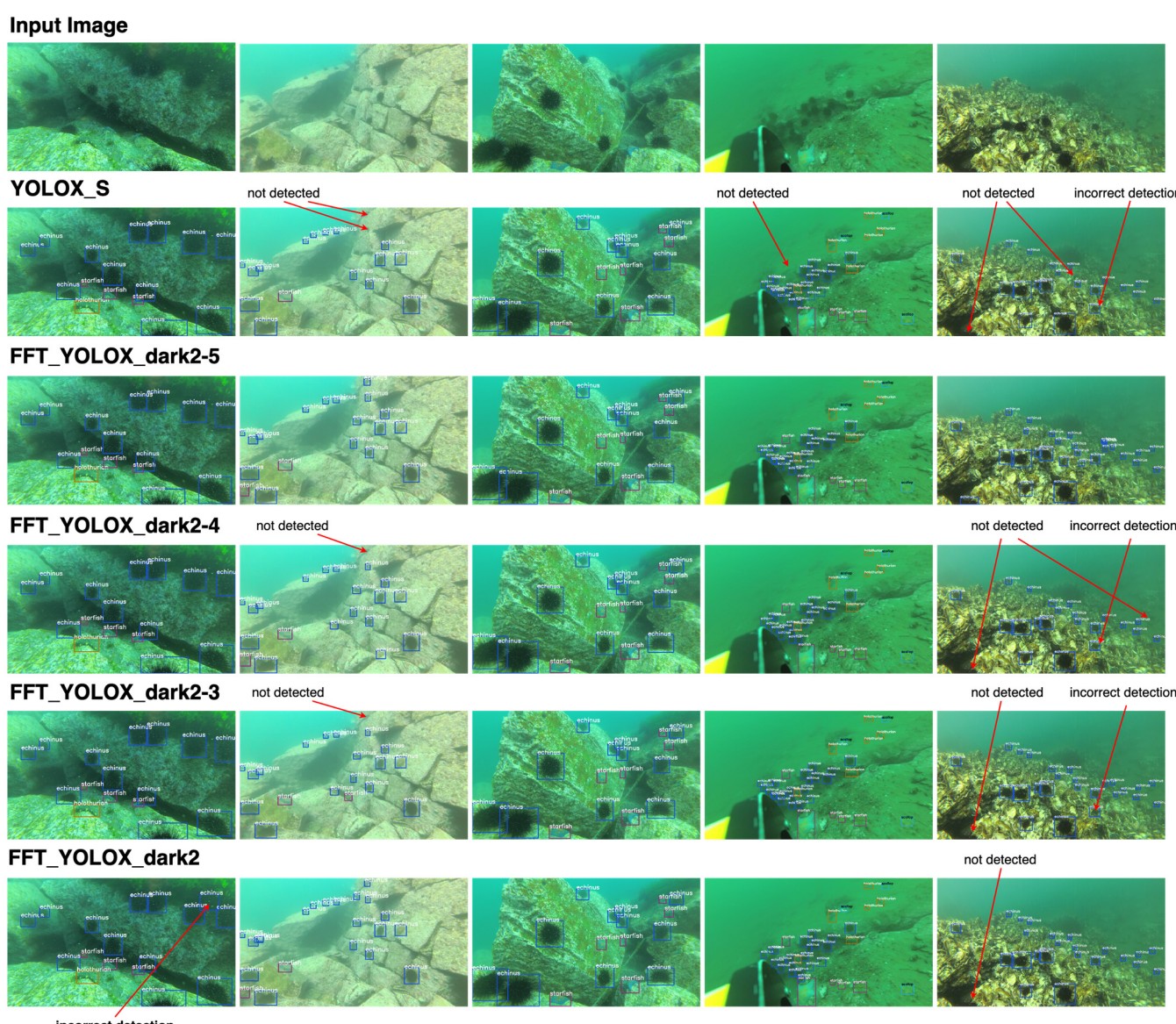

**Figure 5.** The qualitative comparison of the detection results of different models.

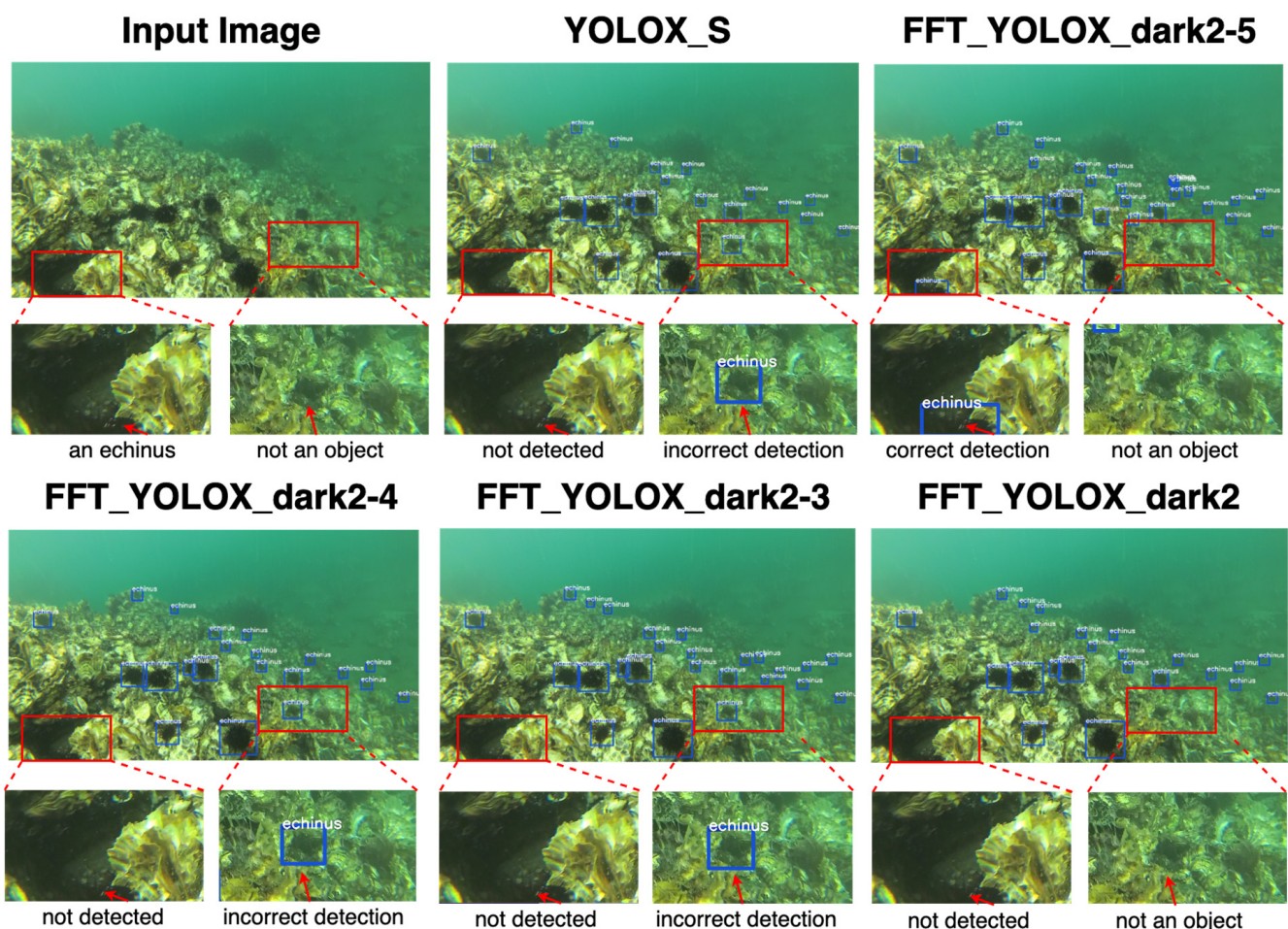

**Figure 6.** The enlarged important areas of the detection results of different models.

## 4. Conclusions

In this paper, a novel detection model FFT_YOLOX based on a modified YOLOX is proposed. Firstly, a unique FFT_Filter was presented, which is a computational efficient and conceptually simple architecture to capture global information of images. Then, a novel FFT_YOLOX model was introduced with fewer model parameters and FLOPs by replacing the standard $3 \times 3$ kernel in the original backbone of YOLOX model with a FFT_Filter for an underwater object detection vision task. This method could quickly and automatically locate the position of a precious marine product, e.g., a sea cucumber, a sea urchin, or a scallop, in this study. Experiments conducted the merged URPC in the context of underwater object detection, demonstrating the effectiveness and generalization of the visual representation of our proposed FFT_YOLOX model. The results of the comparison with the original YOLOX method demonstrated the superiority of the proposed method. The proposed efficient algorithm will help in the image manipulation detection domain and also paves the way for future research in detecting further types of marine precious products.

**Author Contributions:** Conceptualization, P.W. and K.C.; methodology, K.C.; software, P.W.; validation, Z.Y. and H.P.; formal analysis, Z.Y.; investigation, P.W.; resources, K.C.; data curation, T.Z.; writing—original draft preparation, P.W.; writing—review and editing, K.C.; visualization, K.C.; supervision, K.C.; project administration, Z.Y.; funding acquisition, K.C. All authors have read and agreed to the published version of the manuscript.

**Funding:** This work was supported in part by The Foundation of Liaoning Province Education Administration under Grant JL202015, The key R & D projects in Liaoning Province under Grant

2020JH2/10100043 & The Science and Technology Major Special Plan Project of Liaoning Province under Grant 2020JH1/10200002.

**Institutional Review Board Statement:** Not applicable.

**Informed Consent Statement:** Not applicable.

**Data Availability Statement:** The data that support the findings of this study are available from the corresponding author, caikw0602@live.cn, upon reasonable request.

**Conflicts of Interest:** The authors declare that they have no known competing financial interest or personal relationships that could have appeared to influence the work reported in this paper.

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
