# Peer review of "A Novel FFT_YOLOX Model for Underwater Precious Marine Product Detection"

_applsci, doi:10.3390/app12136801_

Round 1
Reviewer 1 Report
To the article has following comments:
- the review can include works on recognition of underwater images of other marine products, not only sea cucumber, sea urchin, the reference of 22 points is not enough for articles in international journals;
- figure 1 with the FFT_YOLOX structure needs to be revised, divided into larger blocks, the text is small, unreadable, background hatching makes it even more difficult to read;
- page 4 contains expressions for converting a map of objects into a spectral region, it is necessary to provide a reference, a textbook, with a description of this theory;
- in section 3. Experiment data set for training taken from the open set (UPRC), why was this particular data set taken?
- figure 5 shows images with different detection methods, but they are hardly distinguishable from each other, it can highlight important areas that are more informative;
- another important factor is illumination, transparency of the underwater environment, how are these factors taken into account? how is the size of seafood in an image based on distance, such as a sea cucumber, determined?
Author Response
Response to Reviewer 1 Comments
Point 1: The review can include works on recognition of underwater images of other marine products, not only sea cucumber, sea urchin, the reference of 22 points is not enough for articles in international journals.
Response 1: Thank you for your valued comments. We have added 12 more publications related to backbones, lightweight models and FFT transform in the “Introduction” and “FFT_filter” sections. Totally, 34 references are included in this paper. We hope it can make the review of this paper be more clear and have enough articles for international journals.
Point 2: Figure 1 with the FFT_YOLOX structure needs to be revised, divided into larger blocks, the text is small, unreadable, background hatching makes it even more difficult to read.
Response 2: Thank you for your valued comments. We have divided the original figure into 2 sub-figures, which are shown in figure 1(a) and figure 1(b). The modified figures have larger blocks and more easier to read.
Point 3: Page 4 contains expressions for converting a map of objects into a spectral region, it is necessary to provide a reference, a textbook, with a description of this theory.
Response 3: Thank you for your valued comments. Following your suggestion, we have provided two FFT references into this work, one is in the introduction part, another is in the section 2.3, which is with a description of the related theory.
Point 4: In section 3. Experiment data set for training taken from the open set (UPRC), why was this particular data set taken?
Response 4: The goal of our work is to propose a practical model for underwater precious marine product detection. However, different from objects detection in openair situation, the underwater image acquisition is much more difficult and expensive. Moreover, the human annotated of precious marine products is labor intensive and time consuming. The UPRC dataset includes sufficient underwater images about pricious marine products with effective label annotations. Therefore, we take the open dataset UPRC to train and evaluate the proposed methods.
Point 5: Figure 5 shows images with different detection methods, but they are hardly distinguishable from each other, it can highlight important areas that are more informative.
Response 5: Thank you for your valued comments. As your helpful suggestion, we have added some red signs into figure 5 to highlight the important areas that are more informative. It can make the detection resutls from different models more distinguishable.
Point 6: Another important factor is illumination, transparency of the underwater environment, how are these factors taken into account? how is the size of seafood in an image based on distance, such as a sea cucumber, determined?
Response 6: To be honest, we did consider the blur problem in an underwater image, and tried some image enhancement methods to improve the image quality. However, the detection results are lower than using the original images. We argue that the traditional image enhancement methods could not help to make better detection performance. In the future work, we would analyze more deep learning based methods to improve the underwater image quality and model performance.
Actually, the size of the precious products in an image is not considered in this work. It not only needs the location in a 2D image plane, but also depth information, which is a distance from a seafood to the camera. In the future work, we intend to acquie an underwater image with a depth camera to obtain both the 2D plane and depth information. Then, a size estimation model can be established and the length even weight of a sea cucumber can be inferenced effectively.

Reviewer 2 Report
The paper by Wang et al. presents an architecture to classify underwater images using a modified convolutional neural network.
The authors provide detailed background analysis and properly justify the selection of the model used.
Although the paper is well written, clear, and flows smoothly, my main concern is, generally speaking, about the overall merit. The innovative content is limited.
Additionally, since the authors want to specifically address three species (sea cucumber, sea urchin, and scallop) through their classification mechanism, a more (maybe) traditional way to present the results would have been to produce AP metrics also separately per each different species.
Apart from such minor remarks, I recommend the publication after some minor corrections.
Some minor typos:
Most of the citations to the references on page 2 don't have a backspace before "[". Please correct.
line 93: "mothed" in place of "method".
line 97: "... and the introduced the FFT filter and ..." unclear, please correct.
line 172: "fig3" in place of "figure 3".
line 175: "Moreover, figure 4 shows that the qualitative comparison results." unclear, please correct.
line 206: "fig 5" in place of "figure 5".
line 209: the title of the table is the one of the template ...
line 228-230: this paragraph is the one of the template ...
Author Response
Point 1: The paper by Wang et al. presents an architecture to classify underwater images using a modified convolutional neural network.
Response 1: Thank you very much for your valued comments.
Point 2: The authors provide detailed background analysis and properly justify the selection of the model used.
Response 2: Thank you very much for your valued comments.
Point 3: Although the paper is well written, clear, and flows smoothly, my main concern is, generally speaking, about the overall merit. The innovative content is limited.
Response 3: Thank you for your valued comments. The goal of this work is to propose a practical model for underwater precious marine products detection. We consider both the size and effective of a detection model. Our innovative content and contribution are to propose a lightweight yet effective precious marine products detection model. Specifically, a unique FFT_filter is presented to replace the standard 3x3 kernel in the original detection model backbone. Finally, some experimental results can evaluate the effectiveness of out proposed model.
In the future work, we would analyze more image enhancement methods to address the blur issue in an underwater image and improve the detection model performance further. Moreover, we intend to acquire an underwater image with a depth camera to obtain both the 2D plane and depth information. Then, a size estimation model can be established and the length even weight of a sea cucumber can be assessed effectively.
We do hope this paper could be considered to publish in this journal. We would take it as an encouragement to continue doing some deeper and more valuable research in this area.
Point 4: Additionally, since the authors want to specifically address three species (sea cucumber, sea urchin, and scallop) through their classification mechanism, a more (maybe) traditional way to present the results would have been to produce AP metrics also separately per each different species.
Response 4: Thank you for your valued comments. As your helpful suggestion, we have added each class AP to evaluate the effectiveness of the proposed method. It is illustrated in table 2.
Point 5: Apart from such minor remarks, I recommend the publication after some minor corrections.
Response 5: Thank you very much. We do wish to be considered for publication in this journal.
Point 6: Some minor typos:
Most of the citations to the references on page 2 don't have a backspace before "[". Please correct.
line 93: "mothed" in place of "method".
line 97: "... and the introduced the FFT filter and ..." unclear, please correct.
line 172: "fig3" in place of "figure 3".
line 175: "Moreover, figure 4 shows that the qualitative comparison results." unclear, please correct.
line 206: "fig 5" in place of "figure 5".
line 209: the title of the table is the one of the template ...
line 228-230: this paragraph is the one of the template ...
Response 6: We are very sorry for our incorrect writing. We have addressed all the mentioned problems in this paper, which can be seen in a “Track Changes” function in a MS Word document.
